# Denuded Descemet's membrane supports human embryonic stem cell-derived retinal pigment epithelial cell culture

Elena Daniele[1,2]*, Lorenzo Bosio[2], Noor Ahmed Hussain[3], Barbara Ferrari[2], Stefano Ferrari[2], Vanessa Barbaro[2], Brian McArdle[4], Nicolò Rassu[5], Marco Mura[1], Francesco Parmeggiani[1,6], Diego Ponzin[2]

1 Department of Translational Medicine, University of Ferrara, Ferrara, Italy, 2 Veneto Eye Bank Foundation, Venice, Italy, 3 Institute of Genetic Medicine, Newcastle University, Newcastle upon Tyne, United Kingdom, 4 The Eye-Bank for Sight Restoration, Inc., New York City, New York, United States of America, 5 Ophthalmic Unit, Ospedale dell'Angelo, Venice, Italy, 6 ERN-EYE Network - Center for Retinitis Pigmentosa of Veneto Region, Camposampiero Hospital, Padua, Italy

* elena.daniele@fbov.it

**Data Availability Statement:** All relevant data are within the manuscript and its Supporting information files.

## Abstract

Recent clinical studies suggest that retinal pigment epithelial (RPE) cell replacement therapy may preserve vision in retinal degenerative diseases. Scaffold-based methods are being tested in ongoing clinical trials for delivering pluripotent-derived RPE cells to the back of the eye. The aim of this study was to investigate human embryonic stem cell-derived retinal pigment epithelial (hESC-RPE) cells survival and behaviour on a decellularized Descemet's Membrane (DM), which may be of clinical relevance in retinal transplantation. DMs were isolated from human donor corneas and treated with thermolysin. The DM surface topology and the efficiency of the denudation method were evaluated by atomic force microscope, scanning electron microscopy and histology. hESC-RPE cells were seeded onto the endothelial-side surface of decellularized DM in order to determine the potential of the membrane to support hESC-RPE cell culture, alongside maintaining their viability. Integrity of the hESC-RPE monolayer was assessed by measuring transepithelial resistance. RPE-specific gene expression and growth factors secretion were assessed to confirm maturation and functionality of the cells over the new substrate. Thermolysin treatment did not affect the integrity of the tissue, thus ensuring a reliable method to standardize the preparation of decellularized DM. 24 hours post-seeding, hESC-RPE cell attachment and initial proliferation rate over the denuded DM were higher than hESC-RPE cells cultured on tissue culture inserts. On the new matrix, hESC-RPE cells succeeded in forming an intact monolayer with mature tight junctions. The resulting cell culture showed characteristic RPE cell morphology and proper protein localization. Gene expression analysis and VEGF secretion demonstrate DM provides supportive scaffolding and inductive properties to enhance hESC-RPE cells maturation. Decellularized DM was shown to be capable of sustaining hESC-RPE cells culture, thus confirming to be potentially a suitable candidate for retinal cell therapy.

**Funding:** This work was partially supported by "5x1000" funds for scientific research from the Italian Ministry of Health and the Italian Ministry of University & Research, and by a Short-Term Scientific Mission grant to ED from the European Cooperation in Science and Technology (EU-COST) Action CA18116 "Aniridia: networking to address an unmet medical, scientific, and societal challenge". There was no additional external funding received for this study. The funders had no role in study design, data collection and analysis, decision to publish, or preparation of the manuscript.

**Competing interests:** The authors have declared that no competing interests exist.

## Introduction

A wide range of disorders is linked to retinal degeneration, generally associated with ultimate loss of vision. Among them, age-related macular degeneration, which is considered the leading cause of visual impairment, myopic macular degeneration and retinitis pigmentosa are the most common retinal degenerative diseases [1–3].

Despite different causes may contribute to the retinal degeneration, the loss of the photoreceptors and the underlying retinal pigment epithelium (RPE) is the common endpoint [4]. In the past, some research groups have demonstrated the possibility of a cell replacement strategy to recover the damaged RPE in the early stages of retinal degeneration [5–8]. These studies highlighted the need of a viable source of RPE cells for successful cell-transplantation therapies. Donor RPE cells from adult human eye, foetus or animals may present unpredictable outcomes when transplanted into the recipient eye. Some negative events are referred to immunological mismatch or functional and/or morphological changes in donor cells. Besides, the number of RPE cells available is not able to meet the high number of patients to be treated. New advances in the field of stem cell therapy pointed towards alternative sources for the generation of retinal cells, including RPE cells [9]. The discovery of human pluripotent stem cells (hPSC), with their self-renewal ability and their potential to differentiate into embryonic tissues, brought to light a reliable tool in the field of regenerative medicine. RPE cells can be efficiently derived from hPSC, including human embryonic stem cells (hESCs) and human induced pluripotent stem cells (hiPSCs).

Current research is now focused on the more suitable method to deliver these cells in the subretinal space. While the injection of an RPE cell suspension may result in poor cell survival and migration from the graft site [10], the transplantation of an RPE monolayer onto a scaffold showed to fulfil retinal transplant requirements, such as regular cell organization, correct integration and photoreceptors support. The identification of a matrix capable of promoting cell adhesion and proliferation should allow the manufacture of a combined construct which is expected to replace the living tissue once implanted and integrated in the correct location of the eye [11].

Although a wide range of both natural and synthetic polymers have been suggested as scaffold, there are still many challenges to face. Despite the easy formulation and manipulation of these matrices, which could also satisfy the large number of patients in need, concerns are addressed to their potential toxicity, mainly due to their inability of recreate the right extracellular matrix (ECM) microenvironment [12, 13].

In these regards, biological scaffolds can represent a simple alternative to other materials. Organization and biochemical composition of membranes isolated from donor tissues proved to be similar to the endogenous Bruch's membrane [14]. Within the eye, these membranes can be found in the milieu of the anterior segment, such as the Bowman's and the Descemet's membranes. Owing to their collagen rich-content, these two layers may offer many advantage features, such as biocompatibility, adhesion and high porosity [15].

In the last decades, new techniques in the field of corneal endothelium transplantation have enabled to retrieve Descemet's membranes from cadaveric donor corneas, thus allowing their re-use for the treatment of several corneal diseases or trauma. The resulting graft after the stripping of the Descemet's Membrane (DM) is a thin sheet of endothelial cells lying upon a collagen matrix that spontaneously rolls itself when it is peeled off from its original location [16].

In this study, we provide evidence that the DM, prior treated to remove any cellular components, may be a reliable substrate for the establishment of a polarized monolayer of hESC-derived RPE cells *in vitro* by mimicking the natural Bruch's membrane.

## Materials and methods

### Preparation of decellularized Descemet's membrane

Cadaveric human corneal specimens were obtained from Eversight (Ann Arbor, MI, USA). Briefly, corneal tissues unsuitable for clinical use were assigned for research purposes with written informed consent signed by the next-of-kin on a 'Donation Disclosure and Acknowledgement' form. Donor corneas were delivered in corneal storage hypothermic medium at 4˚C to the laboratory. Each cornea was placed on a vacuum block and a gentle cut was applied with a 9.5 mm Moria punch (Moria Surgical, Antony, France). The endothelium was stained with Trypan blue, in order to recognize the peripheral edge of the cut. Few drops of phosphate buffered saline (PBS; Thermo Fisher Scientific, Waltham, MA, USA) were added on the top of the endothelium before proceeding with the following step. Using sterile forceps, a gentle stripping of the endothelium was performed from one end to the other with a longitudinal movement. The membrane was then peeled off from its original site, ensuring a complete detachment. Only a small hinge at one peripheral edge was left, thus allowing the labelling of the endothelial side of the membrane with an inverted "F" letter.

For decellularization, the DM was placed in a 15 mL Falcon tube containing a 1 U/mL working solution of thermolysin (Sigma-Aldrich, St. Louis, MO, USA). The bottle was placed at 37˚C for 5 minutes. The treated membrane was then placed in PBS and subjected to fast shaking for 5 more minutes to remove cellular debris. Afterwards, the DM was spread over a 24-well culture plate (Corning, Ney York, NY, USA) or on polyethylene terephthalate (PET) tissue culture (TC) inserts with 1.0 μm pore size, 12-well format (Sarstedt, Nümbrecht, Germany).

DM samples were dried out and terminally sterilized using UV-C irradiation for at least 30 minutes prior to use.

### Decellularization assessment

To evaluate the efficiency of the decellularization method, samples of thermolysin-treated DM were fixed in 4% paraformaldehyde (Santa Cruz Biotechnology Inc., Dallas, TX, USA) at room temperature (RT) for 30 minutes and embedded in Cryobloc compound (Diapath, Bergamo, Italy). Seven μm thick cryosections were permeabilized with 0.1% Triton X-100 (Sigma-Aldrich) in PBS at RT for 30 minutes. The specimens were then blocked in 3% bovine serum albumin (BSA) in PBS at RT for further 30 minutes. Each cryosection was incubated for 1 hour at 37˚C with $Na^+/K^+$ ATPase (1:200; Abcam, Cambridge, UK) and Zona Occludens (ZO-1) (1:200, Thermo Fisher Scientific) antibodies, in order to identify any remaining corneal endothelial cell on the decellularized Descemet's Membrane (dDM) samples. Mouse monoclonal antibodies against collagen type IV α2 chain (1:100) and laminin α5 chain (1:150, both from Merck Millipore, Darmstadt, Germany) were applied to label the structure of the DM. The tissues were mounted on glass slides with DAPI Fluoromount-G (Electron Microscopy Sciences, Hatfield, PA, USA) and examined with Nikon Eclipse Ti-E fluorescent microscope (Nikon, Tokyo, Japan). Images were edited using ImageJ (University of Wisconsin, Wisconsin, USA). Intact samples of corneal endothelium were taken as positive controls.

**AFM analysis.** For atomic force microscopy (AFM) analysis, denuded DMs were rinsed with deionized water. They were spread on top of a thin layer of 2% agarose gel and allowed to dry completely. To avoid any drift of the samples during measurements, they were firmly mounted onto a Superfrost-plus glass slide (Thermo Fisher Scientific) by slightly melting of the gel. Surface topography analysis was performed by static force mode with an Easyscan 2-controlled FlexAFM (Nanosurf, Liestal, Switzerland) equipped with a contact mode cantilever (ContAI-G, Budget Sensors, Sofia, Bulgaria) with a tip radius of 10 nm and nominal spring

constant of 0.2 N/m. The nanostructures of enzymatic-treated DM and intact corneal endothelium were characterized from four randomly regions within each sample, with 256 x two-direction lines scanned at 2 μm/s. Surface roughness was calculated with Mountains SPIP 9 software (Image Metrology A/S, Copenhagen, Denmark) by means of root mean square (Sq) values of the surfaces within selected areas.

**SEM analysis.** The morphology of the decellularized DM was examined using the scanning electron microscope (SEM) FEI Quanta 200 (FEI Company, Hillsboro, OR, USA). A dDM was accommodated on top of a glass slide and allowed to partially dry. The slide was then mounted on stub and studied with the SEM operating at 20 kV.

## Generation of RPE cells from hESCs and culture on decellularized Descemet's membrane

RPE cells were differentiated from hESC line WA09 (WiCell Research Institute, Madison, WI, USA) with a protocol described previously by Hongisto et al. [17]. Briefly, hESCs were dissociated into single cells with TrypLE TM Select Enzyme (Thermo Fisher Scientific). Detached cells were moved to Thermo Scientific™ Nunclon™ Sphera™ Plate, in 15% Knock-Out Serum Replacement (KO-SR) medium, consisting of KnockOut™ Dulbecco's modified Eagle's medium (KO-DMEM) containing 15% Serum Replacement, 2mM GlutaMAX™, 1% MEM Non-Essential Amino Acids, 0.1 mM 2-Mercaptoethanol and 50 U/mL penicillin-streptomycin (all from Thermo Fisher Scientific), supplemented with 10μM Blebbistatin (Sigma-Aldrich), to induce embryoid bodies (EBs) formation. Adherent culture was established after five days of suspension culture. EBs were allowed to spontaneously differentiate in 15% KO-SR medium. After 30 days, pigmented foci were isolated by a surgical blade and digested using TrypLE TM Select Enzyme for ~ 6 minutes at 37˚C. RPE cells were then plated on human recombinant laminin-521 (LN-521; Biolamina, Sundbyberg, Sweden) and collagen IV (Sigma-Aldrich)-coated plates. Once obtained a confluent RPE monolayer, cells were replated on top of acellular DM samples placed inside a 24-well culture plate (Corning) at a density of $1.6 \times 10^5$ cells/cm$^2$. To allow further phenotypic analysis of derived RPE cells, some dDM samples were unfolded over PET TC inserts with 1.0 μm pore size, 12-well format (Sarstedt). Similarly, hESC-RPE cells cultured on LN-521+CIV-coated TC inserts were used as controls, since this formulation of proteins was previously used for the enrichment and the maturation of hESC-RPE cells. Supporting Information include schematic diagram of the differentiation process from hESC and details of RPE cells characterization (S1–S6 Figs).

For pluripotent stem cell cultures, hESC line WA09 (WiCell Research Institute, Madison, WI, USA) was maintained in a feeder-independent culture system on human LN-521 in Essential 8TM Flex Medium (E8; Thermo Fisher Scientific). The genetic profile of the cells was established by the WiCell Research Institute Quality Department (Madison, WI, USA). The Translational Research Initiatives in Pathology Laboratory (University of Wisconsin-Madison, Madison, WI, USA) provided the short tandem repeat (STR) analysis at the request of WiCell Research Institute Quality Department. Fifteen loci plus the Amelogenin gender determining locus were analyzed.

## Adult human RPE cells

Adult human retinal pigment epithelial (ahRPE) cells were obtained from The Eye Bank for Sight Restoration (New York, NY, USA). Primary RPE cells were isolated from one donor with written informed consent for research purpose signed by the next of kin. A cryovial of 5.0 x $10^5$ cells was shipped to the Veneto Eye Bank Foundation in dry ice. After arrival, the cryovial was thawed in a 37˚C water bath for 3 minutes. Thawed cell suspension was centrifuged at

259 x $g$ for 5 minutes. The cell pellet was re-suspended in base medium for ahRPE cells containing DMEM/F12 and αMEM solution with 2% FBS plus additional supplements [18]. For initial cell thawing and plating, base medium was supplemented with FBS to achieve a final concentration of 10% (v/v) FBS. The cells where then plated in Synthemax™ II-SC Substrate (Corning)-coated 24-well culture plate (Corning) at a density of ~ 1.0 x $10^5$/1.9 $cm^2$ and maintained in a humidified incubator kept at 37˚C in 5% $CO_2$. The media was changed every 4th day. Cells were cultured for a minimum of 4 weeks. For gene expression analysis, ahRPE cells were plated either on dDM unfolded over PET TC inserts with 1.0 μm pore size, 12-well format (6.0 x $10^5$ cells/dDM) or Synthemax™ II-coated Sarstedt transwell inserts 1.0 μm pore size, 24-well format (2.0 x $10^4$ cells/insert).

## Cell adhesion and proliferation assays

hESC-RPE cells seeded onto the endothelial-side surface of acellular DM were cultivated for 4 weeks with fresh medium change every 3 days. To test adhesion of viable cells, hESC-RPE cells were incubated in the dark with 1 μM of calcein-acetoxymethyl ester (Calcein-AM) at 24 hours post-seeding for 35 minutes at RT. The cells were visualized with Nikon Eclipse Ti-E fluorescence microscope. ImageJ was used to quantify the number of attached cells at 24 hours. Data were obtained from four independent experiments.

The 5-Bromo-2'-deoxy-uridine Labeling and Detection Kit I (Roche, Indianapolis, IN, USA) was used to detect cell proliferation following the manufacturer's instructions. The proliferation capacity was assessed at 1-week interval. At each time point, the cells were incubated with 10 μM of bromodeoxyuridine (BrdU; Roche) for 45 minutes. After fixation with ethanol for 20 minutes at– 20˚C, the cells were covered with an Anti-BrdU solution (Roche) for 30 minutes at 37˚C, followed by incubation with Anti-mouse-Ig-fluorescein solution (Roche) for 30 minutes at 37˚C and nuclei stained with Hoechst (1:3000; Thermo Fisher Scientific 33342). Cells were then examined using Nikon Eclipse Ti-E fluorescence microscope. Total and positive BrdU cells were quantified from four random visual fields (20X magnification). The data were generated from three independent experiments and all experiments performed in triplicate at each time point.

## Immunofluorescence

After 30 days of culture over dDM, derived RPE cells were permeabilized and blocked as described above for the decellularization assessment. After blocking, the membranes were cut into four approximately equal parts for multiple antibody detection and labeled with primary antibodies against ZO1 (1:200), microphtalmia-associated transcription factor (MITF) (1:100), Bestrophin 1 (Best1) (1:100), orthodenticle homeobox 2 (OTX2) (all from Thermo Fisher Scientific) and $Na^+/K^+$ ATPase (1:200;) in blocking solution overnight at 4˚C. The following day, cells were washed in PBS and incubated with secondary antibodies conjugated with Alexa Fluor 594 or 488 (1:200; Thermo Fisher Scientific) diluted in 3% BSA for 1 hour at RT. Seven μm thick cryosections of hESC-RPE cells over dDM were stained with collagen type IV α2 chain (1:100), laminin α5 chain (1:150) and premelanosome protein (PMEL) (1:50; Thermo Fisher Scientific). All samples were mounted with DAPI Fluoromount-G. Nikon Eclipse Ti-E fluorescent microscope was used to visualize the cells. Z-stack was run to enhance the apical localization of Na+/K+ ATPase protein.

## VEGF enzyme-linked immunosorbent assay

hESC-derived RPE cells over acellular DM and on TC inserts were incubated in serum free media for 48h after 28 and 56 days of culture, respectively. Subsequently, conditioned media

were collected and enzyme-linked immunosorbent assay (ELISA) was carried out to assess the amount of VEGF secreted in the hESC-RPE conditioned culture media with the Human VEGF ELISA kit (Thermo Fisher Scientific) and analyzed at 1:10 dilution according to the manufacturer's instructions.

## Transepithelial resistance (TER)

Transepithelial electrical resistance (TER) was measured along the weeks on hESC-RPE cells cultured over denuded DM placed inside a TC insert with the Millicell® ERS-2 meter and electrode system (Merck Millipore). To obtain true resistance values, TER-values across the samples were subtracted from the measures of denuded DM and empty TC insert without cells. Resulting TER-values ($\Omega$ cm$^2$) were multiplied by the surface area of the TC insert and finally normalized against controls cultured on TC inserts. Measures were carried out in three independent experiments with three samples per condition.

## *In vitro* phagocytosis

hESC-RPE cells cultured for one month over the denuded DM were treated with FITC-labeled latex beads (Sigma-Aldrich) for 48 hours at 37°C. The beads were previously washed in 0.9% sterile NaCl and resuspended 1:100 in 15% KO-SR medium. The cells were then washed three times in PBS and fixed in 4% paraformaldehyde (Santa Cruz) at RT. Cells were then briefly permeabilized with 0.1% Triton X-100 in PBS and blocked with 3% BSA in PBS. Phalloidin (1:1000; Sigma P1951) was used for staining actin filaments. Nuclei were counterstained with Hoechst. The internalization of beads was visualized under Nikon Eclipse Ti-E fluorescent microscope. For quantification, the number of beads ingested was counted using ImageJ. The uptake of beads was expressed as the percentage of cells containing the fluorescence beads.

## Quantitative real-time polymerase chain reaction (qRT-PCR)

Total RNA was extracted from hESC-RPE cells after 30 days of culture on decellularized Descemet's membrane using the RNeasy Plus Mini Kit (Qiagen, Hilden, Germany). The extracted RNA was retrotranscribed into cDNA using the High Capacity cDNA Reverse Transcription kit (Thermo Fisher Scientific). The resulting cDNA was amplified in a quantitative real-time polymerase chain reaction carried out in an Applied Byosistems 7900HT Fast Real-Time PCR system using predesigned TaqMan assays (Thermo Fisher Scientific). The relative levels of gene expression of RPE markers (Bestrophin, RLBP1, RPE65, Tyr, Pmel, MERTK) were determined by the comparative Ct method using the $2^{\Delta\Delta CT}$ and normalised to GAPDH housekeeping gene. Each amplification was run in triplicate and at least three independent experiments were analysed. hESC-RPE cells and primary human RPE cells cultivated on TC inserts were designated as reference samples.

## Statistical analysis

Data are expressed as mean ± SD. Unpaired t-test was performed using GraphPad Prism version 5.0.0 (GraphPad Software, San Diego, CA, USA). Results were considered significant if $p < 0.05$.

## Results

### DM decellularization assessment

Samples of corneal endothelium were analysed before and after the decellularization treatment with a stereoscopic microscope (Fig 1A). Corneal endothelial cells removal was investigated by

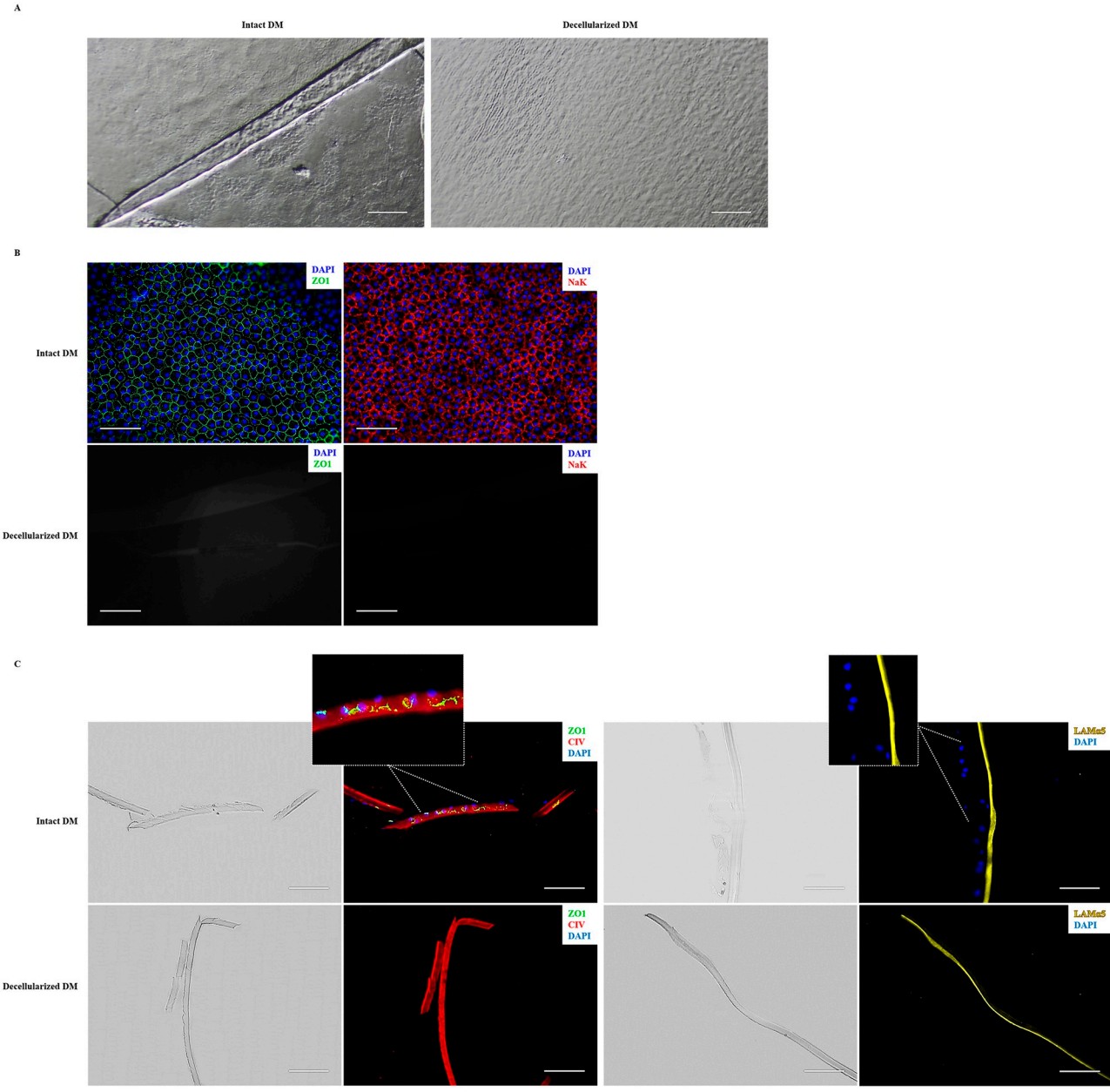

**Fig 1. Assessment of Descemet's Membrane denudation technique.** (**A**) Representative stereoscopic microscope images of the DM before (*left panel*) and after (*right panel*) the decellularization process. (**B**) Immunofluorescence showing the expression of tight junction protein ZO1 (*left panels*; *green*) and sodium-potassium pump NaK (*right panels*; *red*) on DM before (*upper panels*) and after (*lower panels*) the decellularization process. (**C**) Immunostaining of ZO1 (*green*), type IV collagen (*red*) and Laminin α5 (*yellow*) on DM cryosections before (*upper panels*) and after (*lower panels*) the decellularization process. The bright-field images show the entire intact (*upper panels*) and denuded (*lower panels*) tissues. Dotted rectangles indicate the regions that are magnified in the adjacent panels. All nuclei were stained with DAPI (*blue*). Scale bars = 100 μm. DM, Descemet's membrane; NaK, Na$^+$/K$^+$ ATPase.

immunostaining. No endothelial cells were found on thermolysin-treated DM, according to the lack of ZO1 and Na$^+$/K$^+$ ATPase on en face immunofluorescence staining of the tissue (Fig 1B). Laminin α5 and type IV collagen showed positive staining in control and dDM. The expression of these two ECM markers on dDM indicate that the integrity of the membrane was preserved after treatment. Furthermore, the enzymatic treatment did not affect the

polarized pattern expression pattern of these markers, since Laminin α5 was restricted to the endothelial rim of the DM, while type IV collagen was scattered along the stromal side (Fig 1C).

## Surface analysis via AFM and SEM

AFM was used to emphasize any morphological changes occurring after enzymatic treatment. A pre-stripped whole corneal endothelium was chosen as positive control. This tissue had areas free of endothelial cells that allowed DM exposure for the AFM assay. We compared the topography and the surface roughness of these areas with DM underwent decellularization. Controls and thermolysin-treated samples revealed a complex topography made of interspersed twisted fibers forming a densely packed structure. Several bumps and randomly oriented small diameter collagen fibrils were visible across the surface. The basement membrane-like matrix forms a close-knit and uniform design with small pores scattered along the surface. Compared to thermolysin-treated DM, the endothelial-side surface of the control membrane exhibited larger and more evident collagen fibrils (Fig 2).

According to literature, ECM composition and properties have a major impact on cell behaviour [19, 20]. Among others, we estimated surface roughness to underline a possible correlation with cell attachment. Roughness parameters were calculated as mean Sq values for intact and denuded DM samples in four different spots along the endothelial-side of the tissues (Table 1). Surface analysis indicated a decrease in roughness of the decellularized membranes by ~ 50%, which was likely due to a smoothening of the surface following the enzymatic treatment. To provide evidence of the relation between surface roughness and cell adhesion, we calculated the Sq value of the stromal-side of the DM, where hESC-RPE cells were unable to grow. We found a further reduction in roughness values compared to control condition, as a mirror of the different composition of the DM face in front of the corneal stroma.

SEM imaging did not provide additional information about the microstructure morphology of the dDM. The membrane showed a smooth and compact surface, but not appreciable detection of a collagenous meshwork (Fig 3). At higher magnification, images became blurring without distinct features.

## hESC-RPE cells repopulate the decellularized DM

To confirm the ability of the denuded DM to support RPE cell culture, hESC-derived RPE cells were seeded over the endothelial-side surface of the dDM and cultured for one month. At 24 hours, calcein-AM staining was used to calculate the number of adherent cells expressed in relation to the initial cell seeding density (Fig 4A). Compared to control condition on pre-coated wells, higher attachment efficiency was observed for cells on dDM (Fig 4B). BrdU assay was carried out to assess the growth rate of RPE cells along the weeks. The incorporation of the BrdU represented by the percentage of BrdU positive cells was quantified for RPE cells cultured over dDM and compared with the percentage found for control samples (Fig 4C). The percentage of proliferative cells at 24 hours was greater for cells over dDM, demonstrating the affinity of the RPE cells for the new substrate.

Confluence was reached after 1 week in both cultures and no substantial difference was observed in morphology. RPE cells developed their characteristic hexagonal shape and they started to acquire a rich pigmentation from the second week of culture (Fig 5A). The staining pattern of ZO1 revealed consistent morphology and a uniform organization of intercellular junctions of RPE cells after 4 weeks of culture over dDM. In addition, MITF, BEST1 and OTX2 proteins were investigated by immunostaining to display the *in vitro* expression of typical mature RPE markers (Fig 5B). The cryosections shown in Fig 5C enabled the visualisation

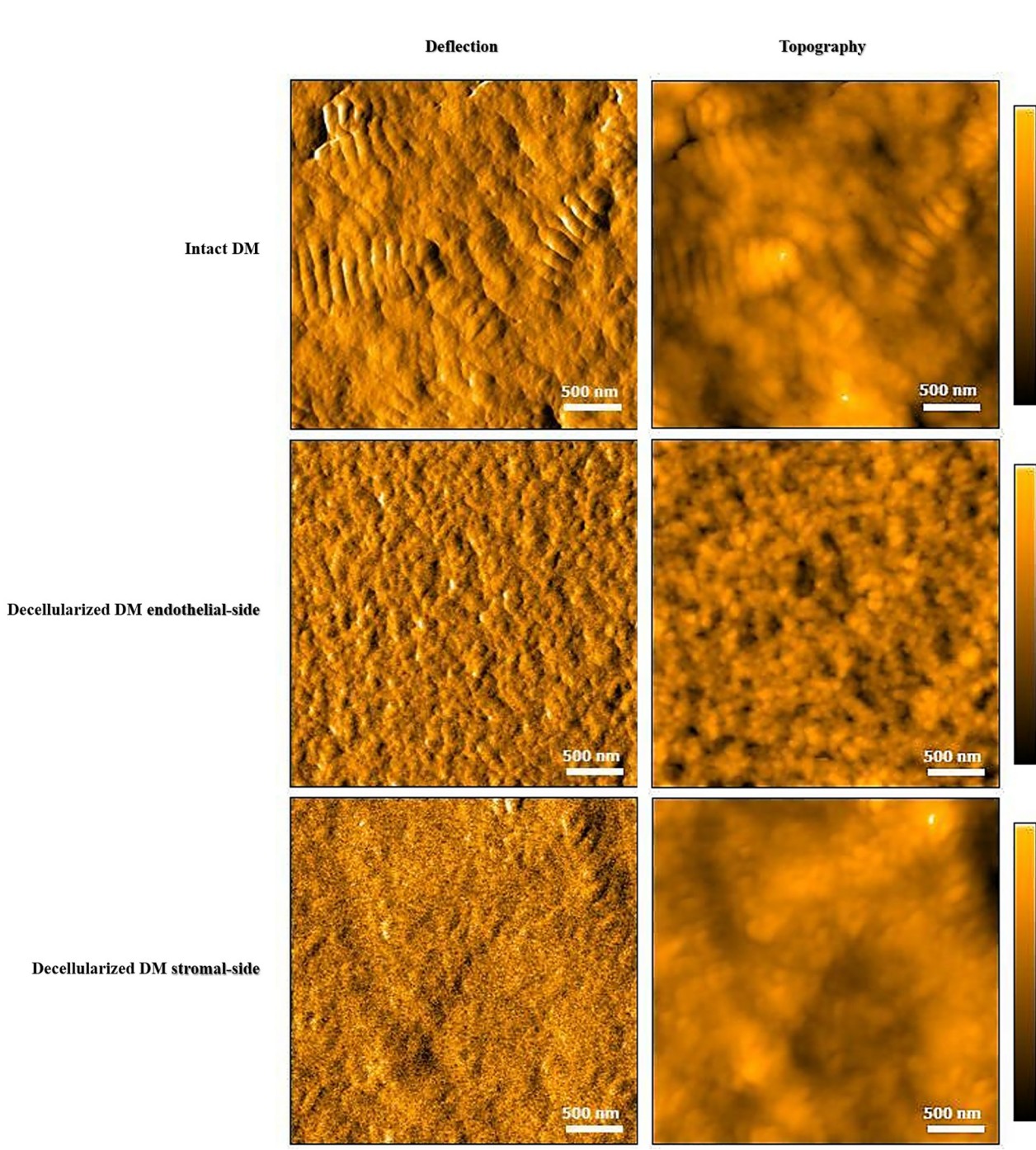

**Fig 2. AFM analysis of Descemet's membrane's surface before and after decellularization.** AFM images representing (**A**) deflection and (**B**) topography of intact and denuded tissue. Image scale: 3 x 3 µm. The coloured bars depict the height. Scale bars = 500nm. DM, Descemet's membrane.

**Table 1. Calculation of roughness.**

| Sample | Roughness [nm] |
|---|---|
| DM | 10.29 ± 3.31 |
| dDM endothelial-side surface | 5.09 ± 0.72 |
| dDM stromal-side surface | 3.14 ± 0.57 |

Roughness of Descemet's membrane, denuded Descemet's membrane endothelial-side surface and denuded Descemet's membrane stromal-side surface samples.

of the hESC-RPE monolayer on top of the denuded DM through PMEL labelling. The tissue was stained with laminin α5 and type IV collagen.

## Characterization of hESC-RPE cells on dDM

Properties and functionality of hESC-RPE cells on the new substrate were further investigated through TER measurements and VEGF secretion. Polarization and growth factors secretion are distinctive features of the RPE cells. Apical localization of the $Na^+/K^+$ ATPase indicated correct polarization of the cell monolayer over the DM (Fig 6A). VEGF secretion occurs baso-laterally, thus supporting the physiology of the choriocapillaris [21]. Using quantitative ELISA, we measured the concentration of VEGF in the culture medium of hESC-RPE cells cultured on dDM after 4 and 8 weeks of culture, respectively. The resulting data were compared with VEGF secretion by hESC-RPE cells cultured on TC inserts, which were expected to release basolaterally VEGF in a time-dependent manner. According to *in vivo* scenario, most of the growth factor was basally secreted in both cultures. We found that VEGF was significantly higher in the basal medium of hESC-RPE cells on dDM than in the related control condition on TC inserts after 28 days of culture and this difference was consistent over time, although not significantly (Fig 6B). Tight junction formation was assessed by measuring TER. The time-dependent rise in resistance demonstrated proper barrier function of the hESC-RPE cell sheet over the new substrate. The mean TER was around 310.9 $\Omega cm^2$, which resulted to be moder-ately lower than the correspondence mean TER of controls (398.4 $\Omega cm^2$). Nevertheless, a decrease in TER values were observed on the control from week 56, which does not occur in the epithelium on the dDM (Fig 6C). Phagocytosis assay showed the ability of hESC-RPE cells

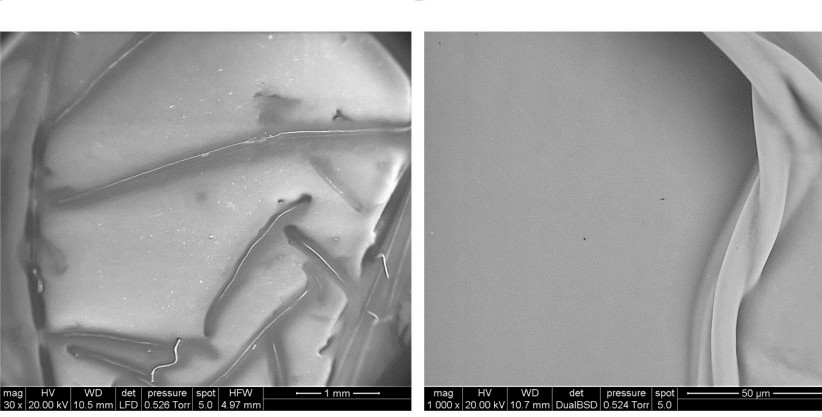

**Fig 3. SEM analysis of decellularized Descemet's membrane's surface morphology.** SEM images of dDM showing a dense membrane with several folds interspersed along the surface: (**A**) X 30, (**B**) X 1000.

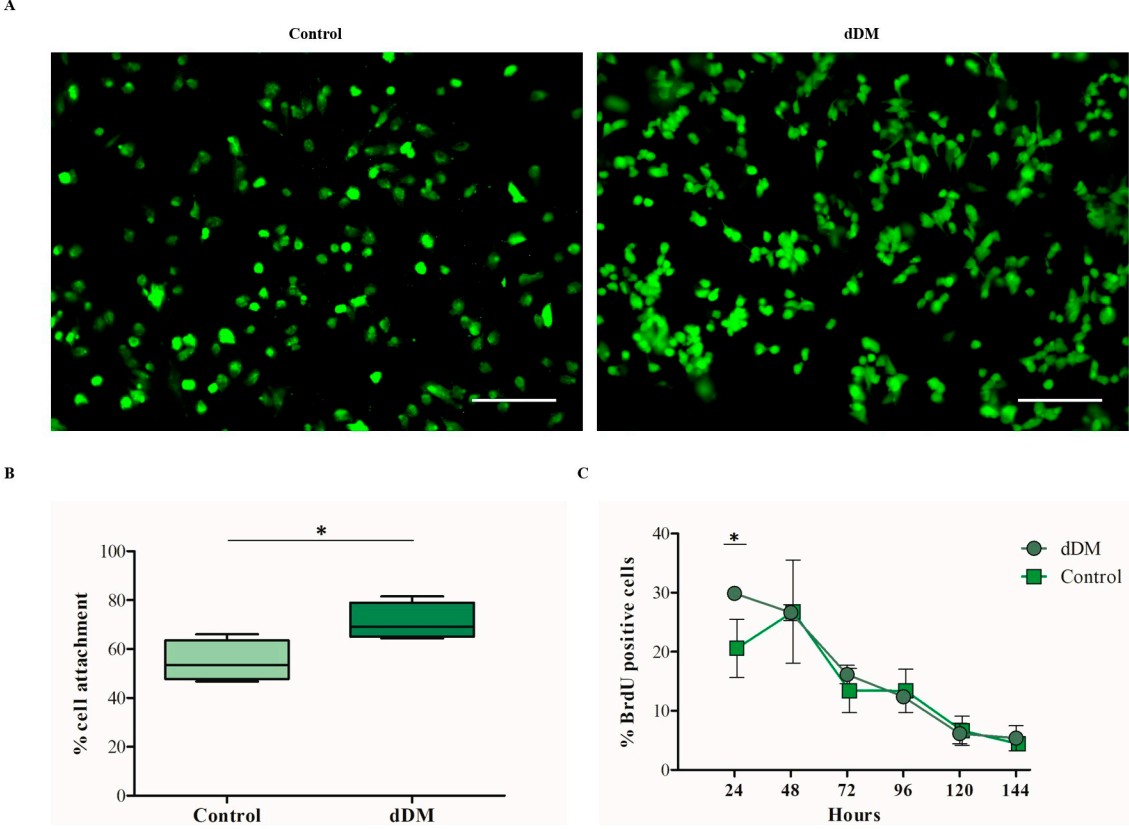

**Fig 4. Decellularized Descemet's membrane provides optimal cues for hESC-RPE cell initial attachment and proliferation. (A)** Immunofluorescence for Calcein AM (*green*) allows the visualisation of viable hESC-RPE cells 24 hours post-seeding on dDM and in control condition. Scale bars = 100 μm. (**B**) Percentage of initial cell attachment on dDM compared to control. Each box represents mean ± s.d. for fourfold experiments. *p = 0.028. (**C**) The graph shows the percentage of proliferative BrdU-positive cells on dDM and on TC insert at different time points during the first week of culture. Each time point represents mean ± s.d. for triplicate experiments. *p = 0.032.

to internalized latex beads when cultured over the dDM, confirming the functionality of these cells over the new strut (Fig 7).

## dDM boosts hESC-RPE cell maturation

The differentiation of the hESC-RPE when cultured on dDM or pre-coated multiwells was assessed by measuring the expression levels of early and late RPE related genes. RNA was collected from differentiated hESC-RPE cells after 4 weeks of culture on dDM. The levels of RLBP1 and MERTK were slightly increased in derived RPE cells cultivated over the Descemet tissue, even if the differential expression pattern did not show statistically significant differences. On the other hand, a significant variation was found in the expression levels of RPE65 (Fig 8A). This result suggests the presence of cues on the dDM that could speed up the maturation of RPE cells derived from pluripotent stem cells. To corroborate these findings, ahRPE cells from donor were tested for gene expression analysis. These cells showed also a significant boost in RPE65 expression level when plated over the dDM compared with the same cell lineage cultured on Synthemax™ II-coated multiwells (Fig 8B). A decrease in the expression of genes involved in the visual cycle is expected in ahRPE cells. In this regard, the significant increase in RPE65 expression during the culture of ahRPE cells on dDM could indicate that

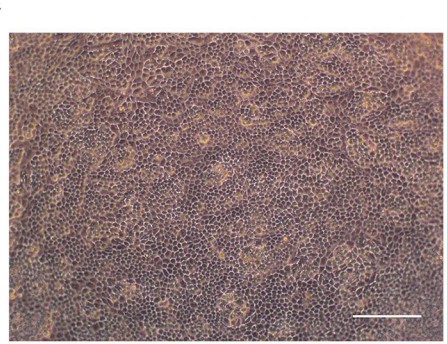

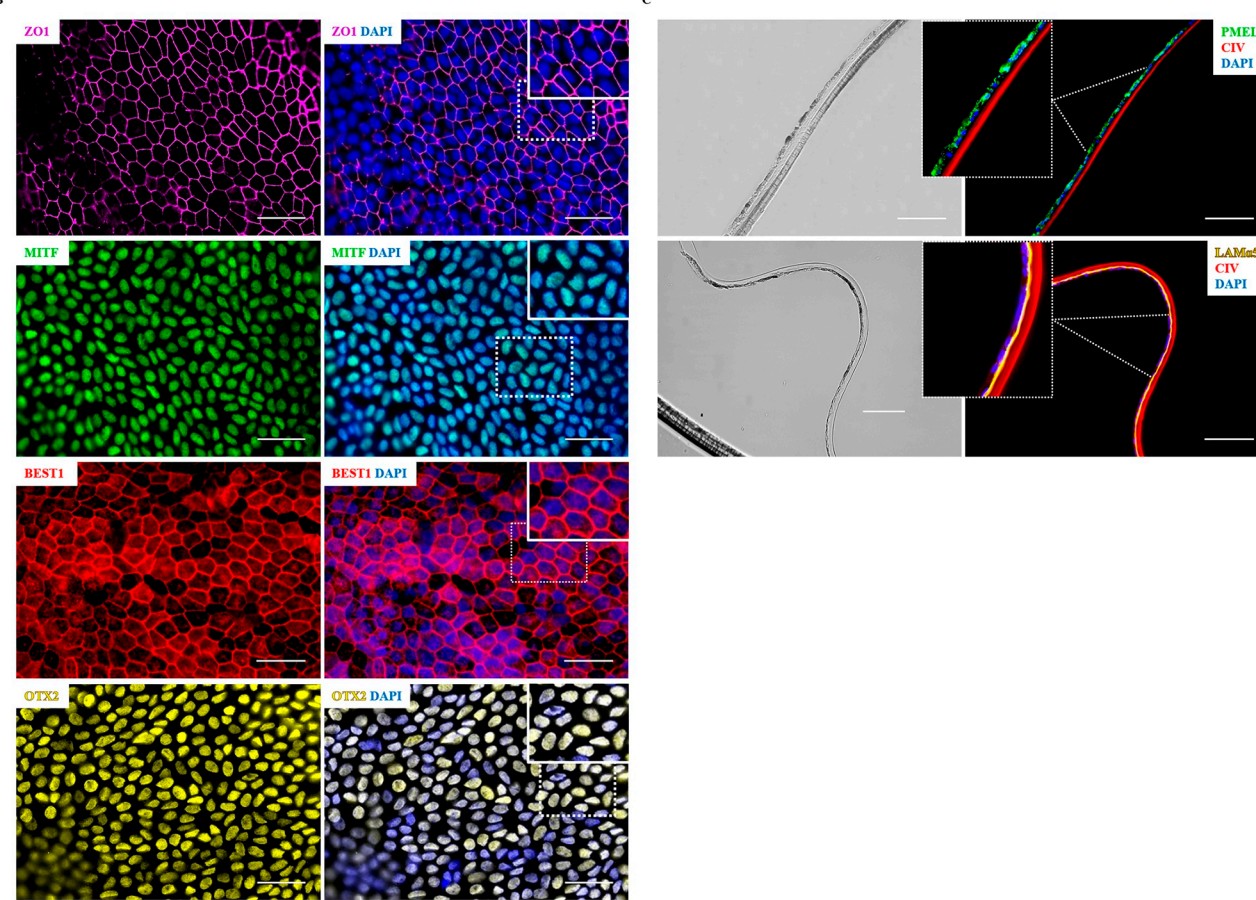

**Fig 5. hESC-RPE cell culture over the denuded Descemet's membrane.** (**A**) Phase-contrast image of hESC-RPE cells cultured for 4 weeks on dDM. Scale bar = 100 μm. (**B**) Immunostaining for typical RPE markers ZO1 (*magenta*), MITF (*green*), BEST1 (*red*) and OTX2 (*yellow*) in hESC-RPE cells cultured on dDM after 1 month of culture. Dotted rectangles indicate the regions that are magnified in the adjacent panels. Scale bars = 100 μm. (**C**) Immunofluorescence of hESC-RPE cells cryosections cultured on dDM after 1 month of culture, demonstrating the development of a confluent monolayer of hESC-RPE cells over the denuded ocular membrane. hESC-RPE cells express premelanosome protein PMEL (*green*). Denuded DM shows positive expression for CIV (*red*) and LAMα5 (*yellow*). Dotted rectangles indicate the regions that are magnified in the adjacent panels. All nuclei shown in blue were counterstained using DAPI. Scale bars = 100 μm. Best1, bestrophin1; CIV, type IV collagen; LAMα5, laminin α5.

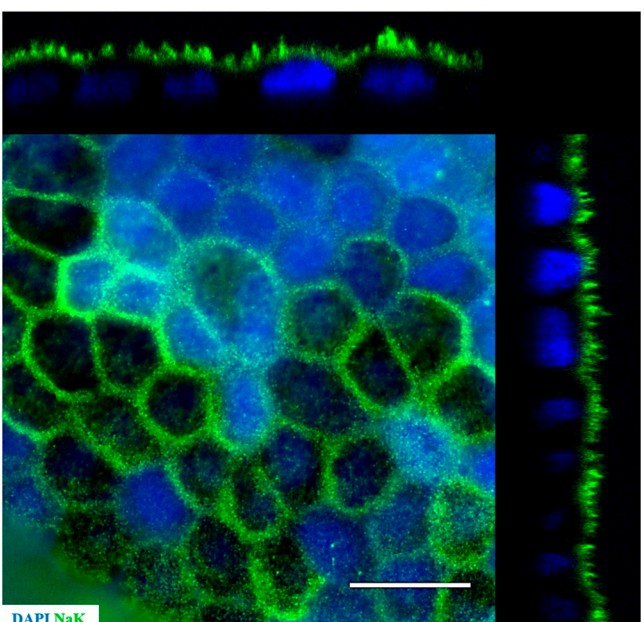

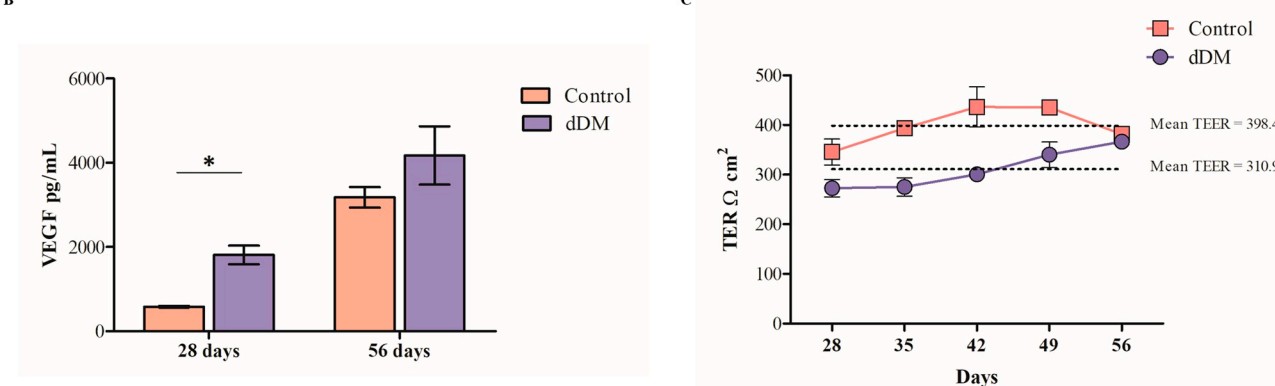

**Fig 6. hESC-RPE cells monolayer evaluation over the natural scaffold.** (**A**) Na$^+$/K$^+$ ATPase immunostaining shows proper polarization of the cells when cultured on dDM. Scale bar = 100 μm. (**B**) ELISA assay was used to quantify VEGF secretion in spent medium from hESC-RPE cells cultured on dDM. VEGF levels were then compared with the amount of VEGF secreted by control on pre-coated TC insert. Values are given as the mean ± s.d. for triplicate experiments. $^*$p = 0.016. (**C**) TER measurements of hESC-RPE cell sheets on dDM at 1 month after seeding, indicating an increase in resistance following proper tight junction development. Each line represents records from four separate cultures of hESC-RPE cells grown on dDM and on TC inserts, respectively. dDM, denuded Descemet's membrane; NaK, Na$^+$/K$^+$ ATPase; TER, transepithelial resistance.

the membrane is able to retain or restore RPE cell function, probably due to its molecular composition, thus providing an ideal microenvironment for hESC-RPE cell culture.

## Discussion

New frontiers in the field of regenerative medicine for ocular diseases include restoration of the cell microenvironment. Transplantation of highly purified pluripotent-derived cells is mandatory but not sufficient to achieve optimal cell therapy outcome. In degenerating retina conditions such as age-related macular degeneration, the presence of an aged or damaged Bruch's membrane can represent a limit at the time of surgery [14], leading to poor graft

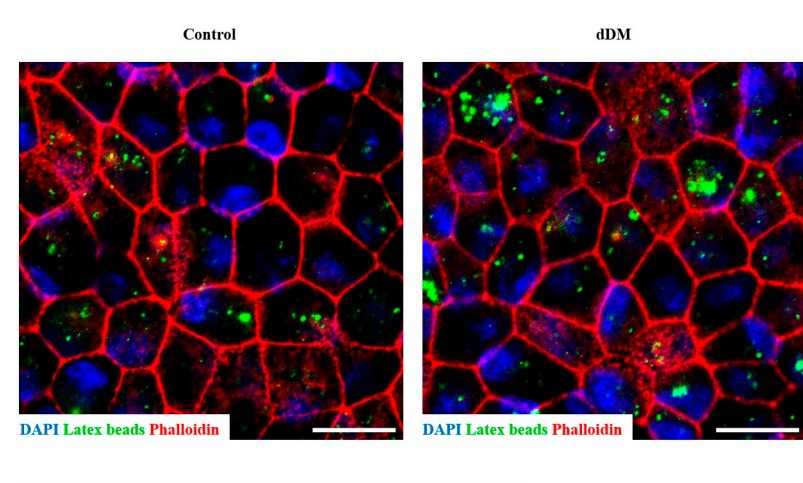

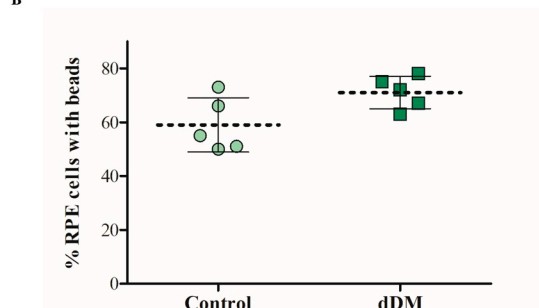

**Fig 7. Phagocytic function of pluripotent-derived RPE cells on dDM.** (**A**) Latex beads uptake (*green*) in hESC-RPE cells over dDM. Actin filaments were stained with phalloidin (*red*). Scale bar = 50 μm. (**B**) Percentage of hESC-RPE cells internalizing the fluorescent beads when cultured on the dDM and under control condition, respectively. Latex beads counts were obtained from five individual field if views, with each field containing ≥ 80 cells. Data represent mean ± s.d. *p = 0.062. dDM, denuded Descemet's membrane.

survival. Recent clinical studies demonstrated the safety of pluripotent-derived RPE cells when implanted as a monolayer grown on a bioengineered scaffold. Despite the encouraging results of this approach in terms of cell polarisation, migration and stability, further efforts need to verify if this approach can provide an ideal cell environment even in extreme RPE cell dysfunction [22]. Although many polymers have been used in the eye because of their biocompatibility (natural polymers) and adaptability (synthetic polymers), human membranes from donor tissues have the invaluable advantage of mimicking natural mechanical properties [23].

The DM is a specialised extracellular matrix underlying the corneal endothelium. Critical for corneal integrity, it is composed of different types of laminin and type IV collagen [24]. In this study we aim to investigate the DM as a substrate for hESC-RPE cells, thus considering this membrane a potential replacement for the Bruch's membrane. Indeed, protein composition and the accessibility of the DM with the modern eye surgical techniques make this tissue a feasible candidate as substitute for the natural RPE basement membrane. After the Descemet stripping, the removal of corneal endothelial cells is performed prior to hESC-RPE culture. Corneal endothelium presents as a monolayer of fully differentiated and static cells [25]. Considering their inability to proliferate, we find that the proposed enzymatic treatment leads to complete and easy removal of endothelial cells, ensuring a reliable protocol to standardise decellularized DM.

Atomic force microscope analysis reveals that the Descemet membrane possesses a rich topography, forming a dense structure of hardly visible small pores. This organisation reflects

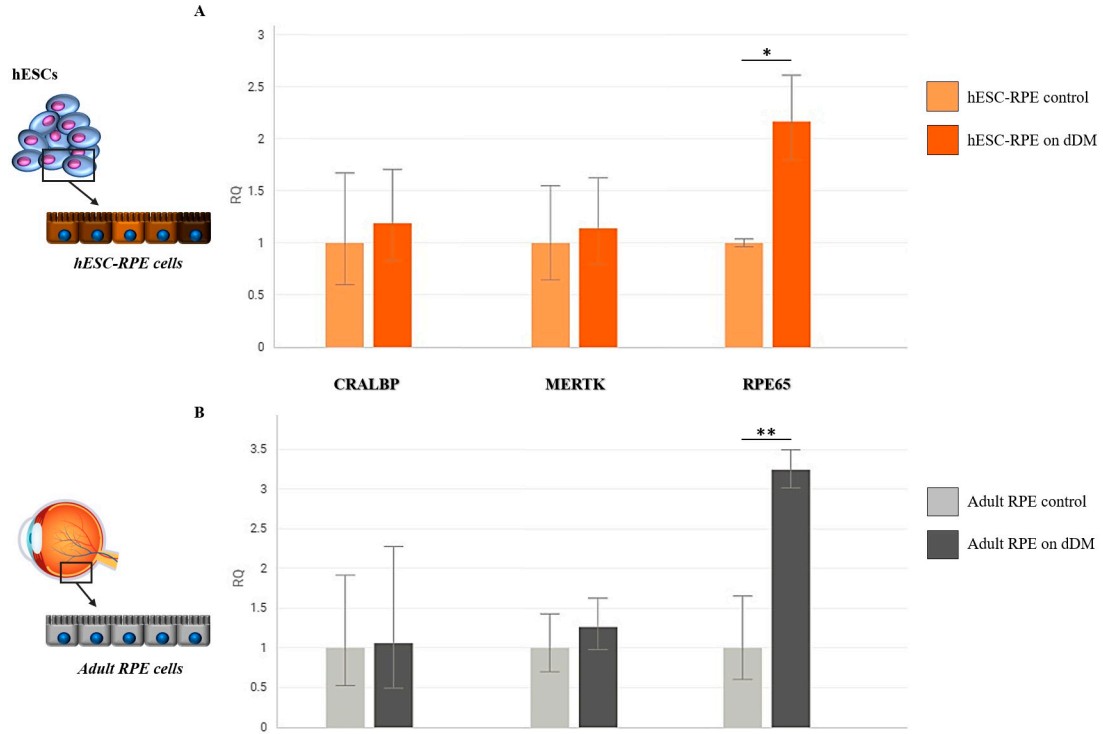

**Fig 8. Quantitative analysis of gene expression of RLBP1, MERTK and RPE65 by real-time PCR.** The bar graph in **A** shows the gene expression comparison between hESC-RPE cells cultured on dDM and hESC-RPE cells cultured on pre-coated TC inserts used as control. In **B**, the same comparison is carried out between adult RPE cells cultured on dDM and adult RPE cells cultured on Synthemax™ II-coated multiwells. Measurements in each graph were normalized to the expression level of related control condition. Values are given as the mean ± s.d. for triplicate experiments. *p = 0.027 and **p = 0.020. hESC, human embryonic stem cells; RPE, retinal pigment epithelium; dDM, denuded Descemet's membrane.

its function within the context of the corneal endothelium. By regulating the corneal water content, the DM acts as a filter, allowing the introduction of small molecules into the aqueous humour [26].

Culture surface can influence cell fate, which in turn responds by adjusting their behaviour [27]. Despite the acceptable biocompatibility of the DM as a natural scaffold, we investigated whether surface properties following decellularization of the membrane could impact on cell adhesion. Surface roughness has been positively correlated with cell adhesion and proliferation. It has been proven that rough surfaces, as well as improving surface texture, can facilitate cell attachment [28]. Our results show a decrease in surface roughness after the treatment of the DM. The enzymatic digestion of thermolysin could be responsible for smoothing the surface of the DM, probably due to removal of some remaining ECM ligands [29] within the posterior non-banded layer of the DM, mainly produced by corneal endothelial cells. Nevertheless, our results encourage the use of the DM for hESC-RPE cells culture, even in the presence of some loss into the ECM framework of the DM endothelial-side after the enzymatic treatment, the latter being undeniable mandatory to ensure a scaffold free of any previous cellular component. On the other hand, the analysis of the stromal side of the Descemet exhibits differences in roughness compared to the endothelial-side, suggesting individual properties of the two sides of the same tissue. The different deposition of biomolecules during DM development and the formation of distinct layers in close contact with separate environments elucidate these differences [30]. According to this, the additional decrease in roughness of the stromal-

side of the DM accounts for adhesion deficiency of the hESC-RPE cells when seeded on this surface. Correct DM labelling is therefore imperative to distinguish the endothelial-side of the DM on which developing a functional monolayer of hESC-RPE cells is possible.

We demonstrate that hESC-RPE cell adhesion to dDM occurs in about 24 hours. hESC-RPE cells deposit on the new matrix as cell patches and then proliferate, ultimately organising in a tight monolayer of epithelial-like cells after 7 days of culture. After this time, cells stop proliferating, consistent with the *in vivo* situation where RPE cells are quiescent and division appears only as a result of disease condition [31]. Additional coating on the top of the dDM is unnecessary, since the only dDM is capable of supporting the growth of the cells. Conversely, hESC-RPE cells spread on pre-coated plastic surface by maintaining a single cell-phenotype. The number of adherent cells on the dDM after 24 hours resulted to be significantly higher than that in the control group, thus revealing not only a viable structural support for the hESC-RPE cell culture, but also a beneficial effect of the DM proteins for the hESC-RPE cell attachment. This justifies the prominent cell proliferation activity during the first 24 hours of culture, leading to an increase of the overall cell proliferation rate over the dDM in relation to control condition at 24 hours.

hESC-RPE cells on dDM display excellent cobblestone-like morphology, showing evidence of tight junctions at 2–3 weeks after seeding. In addition, maturation markers appear more rapidly on cells cultured over the Descemet than those on controls.

Transwell cultures of RPE over dDM are established to provide further evidence of the positive influence of the membrane on RPE physiological functions, such as cellular polarity and transepithelial resistance. The TER values obtained by hESC-RPE monolayers grown over dDM indicate that tight junctions are properly formed. Resistance values $> 200 \, \Omega \, cm^2$ are in accordance with previous finding of hESC-RPE monolayers high TER [32]. The moderately lower TER measurements compared to control conditions can be due to technical limitations, having the membrane not perfectly fitted the entire diameter of the TC insert. To overcome this problem and allowing a proper measurement of the TER, a tissue holder could be used for fastening the dDM and positioning it between two chambers halves of a commercially available chamber [33], so that the dDM becomes the only barrier separating the fluids in each chamber half. Despite this limitation, these data corroborate the maintenance of a functional RPE phenotype *in vitro* when culturing cells over the dDM. Furthermore, the apical localization of $Na^+/K^+$ ATPase justifies the correct polarisation of the cell grown over the dDM. We also notice an increase in basal VEGF secretion after one month of RPE culture over dDM compared to control on pre-coated insert. VEGF secretion is a natural occurring process at the basal membrane RPE level. We believe that the higher VEGF concentration is a consequence of pronounced maturation of RPE cells over the dDM, since the increase of VEGF stimulates cell attachment and differentiation in order to mediate functional angiogenesis [34]. Importantly, the phagocytic activity of hESC-RPE cells was enhanced when the cells where cultured over the dDM compared to the control condition, even though not significantly.

We examine the gene expression patterns of RPE marker genes RLBP1, MERTK and RPE65. No significant differences are observed at the end of the 4-weeks culture period, when the cells appear to be mature enough, suggesting the absence of adverse effects on normal hESC-RPE cells biology. Only the expression of RPE65, which is typically involved in visual pigment recycling and maintenance of photoreceptors, is significantly increased when the cells were cultured over the dDM. These results indicate that dDM contains cues that promote hESC-RPE cell maturation. To further verify these data, ahRPE cells are tested for culture over dDM and gene expression analysis. It is well documented that ahRPE cells in culture lose their pigmentation and acquire a more fibroblastic shape, leading to partial dedifferentiation [35]. In our model, primary RPE cells on dDM regain typical and uniform RPE phenotype by

assembling in a closely packed monolayer of hexagonal cells. Moreover, the new substrate significantly boosts the gene expression of RPE65 compared to cells grown on Synthemax™ II-coated multiwells. This underlines likely inhibition of RPE dedifferentiation, along with providing an ideal microenvironment for hESC-RPE cell culture [35].

This study highlights the feasibility of denuded DM to support the attachment, proliferation and viability of hESC-RPE cells over a 4-week period, potentially acting as a biological dressing in the context of the subretinal space.

DM represents a perfect substitute of a whole ECM by owning all the features of a native tissue's basement membrane. Molecular composition could resemble Bruch's membrane characteristics, with prominent content in type IV collagen, laminin and fibronectin. Especially for type IV collagen, its abundant presence in all layers of adult DM can explain hESC-RPE cells' strong affinity for the membrane. As recently reported, RPE cells preferentially interact with this type of collagen, which is found abundantly in the RPE basal lamina [36]. Besides its biocompatibility and its flexible but robust nature, DM holds the advantage of being already accepted for intraocular use at a regulatory level.

DM could also represent an exclusive scaffold for the xeno-free culture of hESC-RPE cells by avoiding the need of any animal component for initial adhesion and culture maintenance.

Descemet's membrane plays as a sieve in the human cornea, with average pore size of 38 nm which prevents the flux of larger macromolecules [37]. Despite its role in corneal homeostasis, as a replacement of the Bruch's membrane the DM must demonstrate the ability of allowing the entry of nutrients vital for retinal physiology. Future studies will be required in order to verify whether DM may have similar diffusive properties of the Bruch's membrane and how the decellularization process could influence its permeability [38].

The difference in thickness between the DM and Bruch's membrane needs further evaluation. To our knowledge, there is a lack of data in the literature regarding DM transplantation into the subretinal space. For this reason, we have no evidence on membrane tolerability in the new environment and on how this substantial difference in thickness would impact on nutrient transport.

Retinal degenerative diseases such as AMD are usually triggered by several risk factors which lead to oxidative stress. Effective cell transplantation should stand a highly oxidative milieu generated by the increased amount of reactive oxygen species (ROS) [39]. A delivery system capable of replacing damaged RPE cells and endure with ROS levels could represent an efficient therapeutic approach. To improve the survival rate of transplanted cells, a suitable cell carrier should act as a shield against oxidative damage [40]. Although we described the dDM as an excellent scaffold for hESC-RPE cells culture, further studies will need to investigate its ability to deal with pathological conditions. In a recent paper, Krishna L. et al. reported a protective role of another natural scaffold toward oxidative stress. Their results showed that decellularized human amniotic membrane has the ability to scavenge ROS in hyperoxic culture conditions, thus preserving primary RPE cells properties [41].

In our case, the dDM is a basement membrane mainly composed of collagen. Despite the remarkable properties of collagen such as biocompatibility and biodegradability, as well as active regulation of cellular phenotype and cell-ECM interactions, its limited mechanical features could lead to likely inability to tackle degenerative environment. Many groups are thereby addressing the challenge of improving the ROS scavenging ability of collagen-based scaffolds, in order to enhance the delivery of implanted cells under ROS microenvironment. First attempts are to modify the proposed scaffolds with compounds having antioxidant properties such as chitosan, catalase and N-acetyl cysteine, which proved to reduce ROS-induced damage [42–44]. Since oxidative stress is driven by an imbalance between ROS and antioxidant levels in the cells, a previous study has demonstrated that a supplementation of vitamin

D3 alleviated the inflammatory response on primary RPE cells cultivated under hyperoxia-induced oxidative stress conditions. This outcome suggests that combining the cell therapy with an antioxidant treatment could promote the integration of healthy transplanted RPE cells in adverse environments [45].

Overall, our findings propose denuded DM as a tissue engineering tool to support pluripotent-derived RPE cells, hence representing a new therapeutic strategy in the field of regenerative medicine for retinal degeneration or a suitable template for future scaffolds for ocular tissue engineering.

## Supporting information

**S1 Fig. Schematic view with bright-field images of the differentiation process carried out to generate hESC-RPE cells.** hESC, human embryonic stem cell; EB, embryoid bodies; RPE, retinal pigment epithelium; LN-521, recombinant laminin-521; CIV, collagen type IV; E8, Essential 8™ Flex Medium; Blebb, blebbistatin; KO-SR, Knock-out™ serum replacement.
(TIF)

**S2 Fig. Confirmation of a pure culture of differentiated RPE cells.** (A) Immunofluorescence showing lack of staining for pluripotency associated factors such as OCT-3/4 and Nanog (all *green*) in hESC derived-RPE cells (upper panel). hESC WA09 cell line was used for comparison (lower panel). Cell nuclei were stained with DAPI (*blue*). Scale bars = 30 μm. (B) Quantitative polymerase chain reaction revealed no expression of OCT-3/4 and Nanog in hESC-RPE. Data are normalized to hESC WA09 cell line. Data represent mean ± SD of N = 3 experiments for each sample. *** $p < 0.0001$. hESC, human embryonic stem cell; RPE, retinal pigment epithelium.
(TIF)

**S3 Fig. Specific protein expression profiling of hESC-RPE cells.** Representative immunofluorescence staining indicating the correct expression and localization of typical RPE markers in hESC-RPE cells merged with Hoechst nuclear staining (*blue*). Scale bars = 30 μm. Best1, Bestrophin 1; ZO1, zona occludens-1.
(TIF)

**S4 Fig. Barrier properties of differentiated RPE cells.** TER was measured to monitor the barrier function of hESC-RPE cells (A) and adult RPE cells (B) on TC inserts over time (days) using a Millicell volt/ohm meter. Data represent mean ± SD of N = 3 experiments for each sample. TER, transepithelial electrical resistance.
(TIF)

**S5 Fig. Functionality of hESC-RPE cells.** Spent culture media were collected from the apical and the basal chambers of the TC inserts and quantified for PEDF (A-B) and VEGF (C-D) concentration using ELISA assays in hESC-RPE cells (A, C) and adult RPE cells (B, D), respectively. Data represent mean ± SD of N = 3 experiments for each sample.
(TIF)

**S6 Fig. Phagocytosis pf latex beads by RPE cells.** Phagocytic function of hESC-RPE cells and adult RPE cells was tested *in vitro* by challenging the cells with FITC-labeled latex beads. Beads ingestion (*green*) was observed for hESC-RPE cells, as well as for adult RPE cells. Phalloidin staining (*red*) was used to enhance the polygonal shape of the cells. Nuclei were counterstained with Hoechst (*blue*). Scale bars = 25 μm.
(TIF)

**S1 Appendix. Characterization of hESC-RPE cells.**
(DOCX)

**S2 Appendix. Short tandem repeat analysis.**
(PDF)

## Acknowledgments

We gratefully acknowledge Prof. Heli Skottman, Dr. Tanja Ilmarinen, Dr. Heidi Hongisto and Biomedical laboratory technicians Outi Melin and Hanna Pekkanen from the BioMediTech Institute, Faculty of Medicine and Life Sciences, University of Tampere, for sharing their expertise on differentiation and characterization of pluripotent-derived RPE cells and for the critical reading of the manuscript.

## Author Contributions

**Conceptualization:** Elena Daniele.

**Data curation:** Elena Daniele.

**Formal analysis:** Noor Ahmed Hussain.

**Investigation:** Elena Daniele, Noor Ahmed Hussain.

**Methodology:** Elena Daniele, Lorenzo Bosio, Noor Ahmed Hussain, Brian McArdle.

**Project administration:** Elena Daniele, Stefano Ferrari.

**Resources:** Noor Ahmed Hussain, Brian McArdle.

**Supervision:** Elena Daniele, Stefano Ferrari, Marco Mura.

**Visualization:** Barbara Ferrari, Stefano Ferrari, Vanessa Barbaro, Nicolò Rassu, Marco Mura, Francesco Parmeggiani, Diego Ponzin.

**Writing – original draft:** Elena Daniele.

**Writing – review & editing:** Stefano Ferrari, Vanessa Barbaro, Nicolò Rassu, Francesco Parmeggiani, Diego Ponzin.

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
