## [Decision Letter · Decision Letter 0]

6 Nov 2022

PONE-D-22-26643Denuded Descemet's membrane supports human embryonic stem cell-derived retinal pigment epithelial cell culturePLOS ONE

Dear Dr. Daniele,

Thank you for submitting your manuscript to PLOS ONE. After careful consideration, we feel that it has merit but does not fully meet PLOS ONE’s publication criteria as it currently stands. Therefore, we invite you to submit a revised version of the manuscript that addresses the points raised during the review process.

We look forward to receiving your revised manuscript.

Kind regards,

Panayiotis Maghsoudlou

Academic Editor

PLOS ONE

Journal Requirements:

2. For studies reporting research involving human participants, PLOS ONE requires authors to confirm that this specific study was reviewed and approved by an institutional review board (ethics committee) before the study began. Please provide the specific name of the ethics committee/IRB that approved your study, or explain why you did not seek approval in this case.

Once you have amended this/these statement(s) in the Methods section of the manuscript, please add the same text to the “Ethics Statement” field of the submission form (via “Edit Submission”)

"This work was partially supported by “5x1000” funds for scientific research from the Italian Ministry of Health and the Italian Ministry of University & Research, and by a Short-Term Scientific Mission grant to ED from the European Cooperation in Science and Technology (EU-COST) Action CA18116 “Aniridia: networking to address an unmet medical, scientific, and societal challenge”."

"This work was partially supported by “5x1000” funds for scientific research from the Italian Ministry of Health and the Italian Ministry of University & Research, and by a Short-Term Scientific Mission grant to ED from the European Cooperation in Science and Technology (EU-COST) Action CA18116 “Aniridia: networking to address an unmet medical, scientific, and societal challenge”."

"The authors declare to have no potential conflicts of interest."

7. We note that you have included the phrase “data not shown” in your manuscript. Unfortunately, this does not meet our data sharing requirements. PLOS does not permit references to inaccessible data. We require that authors provide all relevant data within the paper, Supporting Information files, or in an acceptable, public repository. Please add a citation to support this phrase or upload the data that corresponds with these findings to a stable repository (such as Figshare or Dryad) and provide and URLs, DOIs, or accession numbers that may be used to access these data. Or, if the data are not a core part of the research being presented in your study, we ask that you remove the phrase that refers to these data.

Reviewers' comments:

Reviewer's Responses to Questions

**Comments to the Author**

1. Is the manuscript technically sound, and do the data support the conclusions?

Reviewer #1: Yes

Reviewer #2: Partly

Reviewer #3: Partly

2. Has the statistical analysis been performed appropriately and rigorously? 

Reviewer #1: Yes

Reviewer #2: Yes

Reviewer #3: Yes

3. Have the authors made all data underlying the findings in their manuscript fully available?

Reviewer #1: Yes

Reviewer #2: Yes

Reviewer #3: Yes

4. Is the manuscript presented in an intelligible fashion and written in standard English?

Reviewer #1: Yes

Reviewer #2: Yes

Reviewer #3: Yes

5. Review Comments to the Author

Reviewer #1: The purpose of this study was to investigate whether decellularized Descemet’s membrane (DM) supports the maturation of hESC-derived RPE cells. DMs were isolated from human corneas, corneal endothelium was peeled off and additional Thermolysin treatment was applied. Decellularization was confirmed by immunohistochemistry and Atomic Force Microscopy (AFM). dDM was dried out on tissue culture plates or PET tissue culture inserts and sterilized by UV. hESC-derived RPE cells attached faster and showed higher initial proliferation when plated on the corneal endothelial side of acellular DM (dDM). RPE on dDM produced significantly more VEGF at 4 weeks than RPE cultured on tissue culture inserts. The transepithelial resistance was lower in RPE cultures on dDM, but reached an equal value on d56. hESC-RPE cultures on dDM were polarized and expressed significantly more RPE65. As a control, the authors also cultured adult eyebank-derived RPE cells which similarly showed a higher dree of maturation and significantly more RPE65 expression on dDM. Thus, the authors conclude that dDM is a suitable substrate for RPE cells.

General comments: The paper is well written, and the studies are well designed. The methods are well described, with listing experimental numbers. The figures are excellent. The results are important because dDM is a natural substrate for RPE cells, could serve as a xeno-free substrate for hESC-RPE, and could help to improve transplantation outcomes.

The authors also discuss the limitations of their study as they have not yet tried to transplant RPE sheets on dDM. It is unclear whether dDM has the same diffusion properties as Bruch’s membrane. This is an important question that needs to be answered because of the RPE role in transporting nutrients for photoreceptors from the choroid.

Specific comments:

A data availability statement is missing from the manuscript and only entered in the manuscript submission form. The authors state that some restrictions will apply, but do not specify which restrictions. The authors state that all relevant data are withing the manuscript and its supporting information files.

Author contributions: The authors should list the specific contributions for each author.

(Minor):

p.14, line 340: “showed” should be “shown”

Reviewer #2: Daniele et al., in the manuscript “Denuded Descemet's membrane supports human embryonic stem cell-derived retinal pigment epithelial cell culture” have shown that denuded DM could be used as a scaffold for potential transplantation of the RPE cells for cell transplantation therapy. However, there few concerns that needs to addressed for strengthening the manuscript.

1. The transplantation of RPE is believed to be one of the primary way to restore the RPE functionality in diseases such as AMD. However, it is also well known in the RPE related diseases the local RPE milieu harbor oxidative stress conditions driven by the dying cells and changes in the molecular components of the milieu. The scaffold that carries the RPE cells should be able to withstand the stress of the oxidative stress so that the cells can hopefully restore the normal functionality. It would be worthwhile if the authors provide some evidence of the role of DM in dealing with oxidative stress environment by experimental findings. The authors may show only the expression of soluble VEGF levels and few qPCR markers to show stability of cells cultured on DM under oxidative stress conditions. Or else the authors may add these points in the discussion section.

2. In Figure 4, the staining for BEST protein is not clear, please replace a better image. Since the IF images are taken at lower magnification, the readers would appreciate an inset with higher magnification.

3. It is unclear the figure of 4(B) and Supplementary figure 3. They seem to be same conditions by different images. In case something is missed please make the changes accordingly.

4. The authors have access to adult primary RPE cells. That is an ideal control to compare and show the physiological property of differentiated cells. Please include the results of the adult RPE cells along with the differentiated RPE cells for transepithelial resistance, VEGF, PEDF production.

5. The authors have missed showing the phagocytic property of the RPE cells which is vital component that gets affected in the RPE disease conditions. Please show the results in comparison with adult primary RPE cells.

Reviewer #3: The purpose of this paper is to use the denuded Descemet's membrane (DM) as a reliable substrate for establishing a polarized RPE monolayer for subsequent retinal implantation. Thus, this system might serve as a substitute for Bruch’s membrane (BM).

The denuded DM (dDM) has been used in the past as a substrate mostly for endothelial cells, but there are also have been prior reports of using the DM as a substrate for RPE cells, similar to the main tenet of this paper. In that sense, the paper is not entirely innovative.

The authors characterized their DM-RPE system using a large body of careful experimental approaches, (including consideration of statistics, i.e. replications and sample sizes) and demonstrated that they indeed can decellularize (denude) the DM, repopulate its endothelial side with different kinds of RPE cells derived from hESC-, adult human-, and human iPSC (no justification was provided for the use of these three kinds of RPE cells) and demonstrated that, by comparison to their controls, the dDM has beneficial effects on some, but not all of the RPE cells tissue-specific features.

Criticisms:

1. Confounding the authors' claim of superiority of the dDM is the lack of effect of some of the RPE-specific markers of cell maturation (such as CRALBP and MERTK) as well as the results of their TEER measurements, in which control cells have essentially > 25% higher TEER values than the cells growing on the dDM.

2. The characterization of the dDM by AFM tells something about the surface roughness but has little information as to the surface structure, e.g. morphology (e.g. fibrous)

3. The main problem I have with this paper is that in my opinion, the authors used the wrong controls. For the most part, their controls were RPE cells seeded onto either LN-521+CIV- precoated plastic wells or pre-coated TC inserts. None of these controls properly mimics the 3D structure/microtopography of the dDM. Instead, the authors should have maintained the 3D configuration, e.g. by using micro-/nanofibrous mats electrospun of collagen I (and then applying their LN-521+CIV-coating) or using a combination of ECM proteins as found in the natural subendothelial space in the DM. Furthermore, given that it might be impossible to obtain DMs in sufficient quantities to carry out large-scale clinical studies, the use of a more complex “man-made” substrate might reveal whether the EDM is indeed superior, and/or whether this “synthetic” substrate might suffice to obtain a functional RPE layer ready for implantation.

6. PLOS authors have the option to publish the peer review history of their article (what does this mean?). If published, this will include your full peer review and any attached files.

Reviewer #1: No

Reviewer #2: **Yes: **Debashish Das

Reviewer #3: **Yes: **Peter I. Lelkes, Ph.D.

---

## [Author Response · Author response to Decision Letter 0]

19 Jan 2023

Manuscript ID: PONE-D-22-26643

Response to Reviewers

Dear Prof. Panayiotis Maghsoudlou, 

Academic Editor of PLOS ONE,

Thank you for giving us the opportunity to submit a revised draft of our manuscript titled “Denuded Descemet's membrane supports human embryonic stem cell-derived retinal pigment epithelial cell culture” for publication in PLOS ONE. We appreciate the time and effort that you and the reviewers have dedicated to providing your valuable feedback on our manuscript and we are grateful to the reviewers for their insightful comments and valuable improvements to our paper. We have incorporated most of the suggestions provided by the reviewers, with appropriate references when needed. We have highlighted the changes within the manuscript. Please see below a point-by-point response to the reviewer’s comments and concerns. 

Comments from Reviewer 1:

General comments: The paper is well written, and the studies are well designed. The methods are well described, with listing experimental numbers. The figures are excellent. The results are important because dDM is a natural substrate for RPE cells, could serve as a xeno-free substrate for hESC-RPE and could help to improve transplantation outcomes.

The authors also discuss the limitations of their study as they have not yet tried to transplant RPE sheets on dDM. It is unclear whether dDM has the same diffusion properties as Bruch’s membrane. This is an important question that needs to be answered because of the RPE role in transporting nutrients for photoreceptors from the choroid. 

Response: We thank and appreciate the positive feedback from the Reviewer. We also agree about the next steps that need to be done. It is therefore of utter importance to move forward with permeability studies to see whether the Descemet’s membrane can fully recapitulate Bruch’s membrane properties, including the ability to support sufficient nutrients transportation.

Specific comments: A data availability statement is missing from the manuscript and only entered in the manuscript submission form. The authors state that some restrictions will apply, but do not specify which restrictions. The authors state that all relevant data are within the manuscript and its supporting information files.

Response: We thank the Reviewer for pointing this out. We have decided to change the Data Availability Statement. Indeed, the data can be publicly available without any restrictions applied. None of the provided information could compromise the privacy of the research participants. They are therefore available within the article and its supplementary materials. 

Author contributions: The authors should list the specific contributions for each author.

Response: Agree. We have added Author Contributions on page 25, lines 608 - 618.

(Minor): p.14, line 340: “showed” should be “shown”.

Response: We are sorry for this inattention. We have revised the text accordingly. 

Comments from Reviewer 2:

Comment 1: The transplantation of RPE is believed to be one of the primary ways to restore the RPE functionality in diseases such as AMD. However, it is also well known in the RPE related diseases the local RPE milieu harbour oxidative stress conditions driven by the dying cells and changes in the molecular components of the milieu. The scaffold that carries the RPE cells should be able to withstand the stress of the oxidative stress so that the cells can hopefully restore the normal functionality. It would be worthwhile if the authors provide some evidence of the role of DM in dealing with oxidative stress environment by experimental findings. The authors may show only the expression of soluble VEGF levels and few qPCR markers to show stability of cells cultured on DM under oxidative stress conditions. Or else the authors may add these points in the discussion section.

Response: We would like to thank the Reviewer for raising this very interesting point. Studies on the capability of the dDM, as well as other collagen-based scaffolds to hinder oxidative stress are in need. We think this topic deserves a designated study conducted to enlighten how dDM could withstand these conditions and it could be the focus of our next paper. Therefore, we embraced the Reviewer’s advice and we have discussed this point as a future improvement of our current paper on page 24, lines 572 – 595:

“Retinal degenerative diseases such as AMD are usually triggered by several risk factors which lead to oxidative stress. Effective cell transplantation should stand a highly oxidative milieu generated by the increased amount of reactive oxygen species (ROS) [39]. A delivery system capable of replacing damaged RPE cells and endure with ROS levels could represent an efficient therapeutic approach. To improve the survival rate of transplanted cells, a suitable cell carrier should act as a shield against oxidative damage [40]. Although we described the dDM as an excellent scaffold for hESC-RPE cells culture, further studies will need to investigate its ability to deal with pathological conditions. In a recent paper, Krishna L. et al. reported a protective role of another natural scaffold toward oxidative stress. Their results showed that decellularized human amniotic membrane has the ability to scavenge ROS in hyperoxic culture conditions, thus preserving primary RPE cells properties [41]. 

In our case, the dDM is a basement membrane mainly composed of collagen. Despite the remarkable properties of collagen such as biocompatibility and biodegradability, as well as active regulation of cellular phenotype and cell-ECM interactions, its limited mechanical features could lead to likely inability to tackle degenerative environment. Many groups are thereby addressing the challenge of improving the ROS scavenging ability of collagen-based scaffolds, in order to enhance the delivery of implanted cells under ROS microenvironment. First attempts are to modify the proposed scaffolds with compounds having antioxidant properties such as chitosan, catalase and N-acetyl cysteine, which proved to reduce ROS-induced damage [42-44]. 

Since oxidative stress is driven by an imbalance between ROS and antioxidant levels in the cells, a previous study has demonstrated that a supplementation of vitamin D3 alleviated the inflammatory response on primary RPE cells cultivated under hyperoxia-induced oxidative stress conditions. This outcome suggests that combining the cell therapy with an antioxidant treatment could promote the integration of healthy transplanted RPE cells in adverse environments [45].” 

Please find the new references in the reference list of the main manuscript.

Comment 2: In Figure 4, the staining for BEST protein is not clear, please replace a better image. Since the IF images are taken at lower magnification, the readers would appreciate an inset with higher magnification.

Response: We thank the Reviewer for these valuable suggestions. We have now revised Figure 4B (now Figure 5B), which should now allow a better appreciation of BEST1 staining. Furthermore, additional insets have been incorporated alongside each image within Figure 5B. The appropriate amendments have been reported on the legend of Figure 5 (page 16, lines 378 - 388).

Comment 3: It is unclear the figure of 4(B) and Supplementary figure 3. They seem to be same conditions by different images. In case something is missed please make the changes accordingly.

Response: We understand the Reviewer’s concern on the different images. Figure 4B (now Figure 5B), includes representative images of hESC-RPE cells cultured over the dDM, which show expression of RPE related markers. On the other hand, supplementary figure 3 adds further evidence of correct protein expression by differentiated hESC-RPE cells after the spontaneous differentiation process. 

The Descemet’s membrane allows us to perform optimal immunofluorescence assays without particular issues such as autofluorescence or others. We agree that the two conditions might appear very similar. For this reason, we paired the immunostaining of hESC-RPE cells on dDM with cryosections, in order to highlight the membrane underlying the cells. 

 Furthermore, we assert that the presence of the dDM made high magnification imaging challenging, compared with imaging the control condition (Supplementary figure 3). Indeed, when the membrane is properly stretched over a surface, cell imaging can be performed on the same focal plane at low magnification. However, because the dDM has a natural curved shape as in the cornea, it is difficult to obtain a whole flat structure. This affects high magnification imaging, by making the image appearing on different focal plans. 

Comment 4: The authors have access to adult primary RPE cells. That is an ideal control to compare and show the physiological property of differentiated cells. Please include the results of the adult RPE cells along with the differentiated RPE cells for transepithelial resistance, VEGF, PEDF production.

Response: We thank the Reviewer for this recommendation. The suggested content has been introduced as Supplemental Material to compare the differentiated cells with our line of adult RPE cells. Data on transepithelial resistance, VEGF and PEDF production from adult RPE cells have been included on Figures S4 and S5, respectively. 

Comment 5: The authors have missed showing the phagocytic property of the RPE cells which is vital component that gets affected in the RPE disease conditions. Please show the results in comparison with adult primary RPE cells.

Response: Thank you for pointing this shortcoming out. We have now shown the phagocytic activity of the hESC-RPE cells on TC inserts as well as on dDM on Figure 7. The protocol of the phagocytosis assay has been described on page 11, lines 264 - 273, as well as on S1 Appendix. The results of the assay have been reported on page 17, lines 408 – 409 and on page 22, lines 531 – 533. 

The phagocytic property of adult RPE cells has been exhibited on S6 Fig, along with the phagocytosis of hESC-RPE cells. The corresponding figure legend can be found on page 32, lines 784 – 788.

Comments from Reviewer 3:

Comment 1: Confounding the authors' claim of superiority of the dDM is the lack of effect of some of the RPE-specific markers of cell maturation (such as CRALBP and MERTK) as well as the results of their TEER measurements, in which control cells have essentially > 25% higher TEER values than the cells growing on the dDM.

Response: We thank the reviewer for outlining these valid points. Regarding the RPE-specific markers of cell maturation, the gene expression levels resulting when culturing hESC-RPE cells and adult RPE cells could be ascribed to the in vitro culture itself. hESC-RPE cells express important biochemical RPE markers such as RPE65 and CRALBP. During in vitro culture and differentiation, a downregulation of these genes is expected, since hESC-RPE cells do not communicate with the neural retina. As for the adult human RPE cells, it is well known that these cells undergo dedifferentiation when cultured, therefore decreasing progressively the expression of RPE-related genes during the culture period. These genes will have a different decrease in expression. It has been demonstrated that CRALBP and MERTK show minimal changes of their expression levels during development, while a large decrease in expression is observed for the RPE65 gene, which is involved in the conversion of all-trans retinol to 11-cis retinal and in visual pigment regeneration. Assuming our dDM can create conditions that reduce the likelihood RPE will change, fostering the maintenance of RPE65 and other RPE genes expression during culture. Considering these events, the significant difference observed for RPE65 gene expression could be due to a higher RPE65 downregulation in control conditions compared to the other two genes. The difference in RPE65 gene expression level between RPE cells (both hESC-RPE cells and adult RPE cells) cultured over the dDM and the related controls will result therefore significant. 

We certainly agree with the reviewer about the results of TEER measurements in control cells compared to cells cultured over the dDM, even though the existing difference was not significant. Unfortunately, as we state in the discussion of the paper on page 20 lines 501 - 503, this outcome is due to technical limitation, since the DM does not perfectly fit the entire diameter of the TC insert, thus leaving some empty space on the insert not covered by cells. We are aware this is a downside of our method and we should establish a different system which would allow us to perform the phenotypic analysis more thoroughly. We have therefore presented a possible solution on pages 21 and 22, lines 520 - 523. We now say the following:

“To overcome this problem and allowing a proper measurement of the TER, a tissue holder could be used for fastening the dDM and positioning it between two chambers halves of a commercially available chamber [33], so that the dDM becomes the only barrier separating the fluids in each chamber half.”

Please find the new reference in the reference list of the main manuscript.

Comment 2: The characterization of the dDM by AFM tells something about the surface roughness but has little information as to the surface structure, e.g. morphology (e.g. fibrous).

Response: The comment was really relevant and the surface morphology of the dDM was analyzed using a Scanning electron microscopy. A concise description of the instrument and imaging procedure has been added within the Materials and methods section (page 7, lines 162 - 166). Unfortunately, little information on morphology was obtained from this analysis, since the dDM turned out to be a very dense and thick structure. The outcome of this analysis has been reported on the Results section (page 14, lines 336 - 339), together with the AFM analysis. The new figure 3 shows SEM analysis of a decellularized DM. Corresponding figure 3 caption can be found on page 14, lines 343 – 345.

Comment 3: The main problem I have with this paper is that in my opinion, the authors used the wrong controls. For the most part, their controls were RPE cells seeded onto either LN-521+CIV- precoated plastic wells or pre-coated TC inserts. None of these controls properly mimics the 3D structure/microtopography of the dDM. Instead, the authors should have maintained the 3D configuration, e.g. by using micro-/nanofibrous mats electrospun of collagen I (and then applying their LN-521+CIV-coating) or using a combination of ECM proteins as found in the natural subendothelial space in the DM. Furthermore, given that it might be impossible to obtain DMs in sufficient quantities to carry out large-scale clinical studies, the use of a more complex “man-made” substrate might reveal whether the EDM is indeed superior, and/or whether this “synthetic” substrate might suffice to obtain a functional RPE layer ready for implantation.

Response: We understand the Reviewer’s concern about proper controls. It would have been interesting to compare the dDM with another scaffold mimicking its 3D structure. However, in the case of our study, our primary aim was to evaluate the quality of the hESC-RPE cells culture over a substrate other than that used for their culture in vitro and the ability of this scaffold to preserve their cell features and sustain their functionality. Considering there is no availability yet of a gold standard scaffold for the pluripotent-derived RPE cells culture, we cannot fully estimate and validate the resulting RPE culture on a potential 3D matrix. Therefore, commercial cell culture inserts 1µm pore size coated with collagen IV and laminin were selected as control condition in cell culture studies as we have previously used these proteins for the enrichment and maturation of our hESC-RPE cell. We have added this information on materials and methods on page 8, lines 186 - 188:

 “Similarly, hESC-RPE cells cultured on LN-521+CIV-coated TC inserts were used as controls, since this formulation of proteins was previously used for the enrichment and the maturation of hESC-RPE cells.”

Regarding the likely insufficient number of available DMs, we believe this is an important consideration and we wish to respectfully differ. According to our Eye Bank statistics, more than one-third of the donor corneas we collect every year are deemed unsuitable for corneal transplantation due to the low cellular content of the corneal endothelium. Since our purpose involves a decellularization of the corneal endothelium, these discarded tissues can be recovered and used to improve the transplantation of other ocular tissues such as the RPE. Our ambition is to avoid wasting what is given from donors. 

In addition, we hope this work might encourage other researchers working in the field of tissue engineering to create novel biomaterials with similar qualities to those of the Descemet’s membrane. This membrane holds excellent scaffolding properties for RPE cells and recapitulates the environment of a late organ development. We have now suggested this potential use of the DM at the end of the article on page 25, lines 598 - 599:

 “…or a suitable template for future scaffolds for ocular tissue engineering.”

---

## [Editor Report · Decision Letter 1]

23 Jan 2023

Denuded Descemet's membrane supports human embryonic stem cell-derived retinal pigment epithelial cell culture

PONE-D-22-26643R1

Dear Dr. Daniele,

We’re pleased to inform you that your manuscript has been judged scientifically suitable for publication and will be formally accepted for publication once it meets all outstanding technical requirements.

Kind regards,

Panayiotis Maghsoudlou

Academic Editor

PLOS ONE

---

## [Editor Report · Acceptance letter]

27 Jan 2023

PONE-D-22-26643R1 

Denuded Descemet’s membrane supports human embryonic stem cell-derived retinal pigment epithelial cell culture 

Dear Dr. Daniele:

I'm pleased to inform you that your manuscript has been deemed suitable for publication in PLOS ONE. Congratulations! Your manuscript is now with our production department. 

Kind regards, 

on behalf of

Dr. Panayiotis Maghsoudlou 

Academic Editor

PLOS ONE